# A Nickel-Containing Polyoxomolybdate as an Efficient Antibacterial Agent for Water Treatment

**DOI:** 10.3390/ijms23179651

**Published:** 2022-08-25

**Authors:** Jiangnan Chang, Mingxue Li, Jiyuan Du, Min Ma, Cuili Xing, Lin Sun, Pengtao Ma

**Affiliations:** Henan Key Laboratory of Polyoxometalate Chemistry, College of Chemistry and Chemical Engineering, Henan University, Kaifeng 475004, China

**Keywords:** polyoxomolybdate, antibacterial activity, antibacterial mechanism, adsorption properties

## Abstract

In view of the water pollution issues caused by pathogenic microorganisms and harmful organic contaminants, nontoxic, environmentally friendly, and efficient antimicrobial agents are urgently required. Herein, a nickel-based Keggin polyoxomolybdate [Ni(L)(HL)]_2_H[PMo_12_O_40_] 4H_2_O (**1**, HL = 2-acetylpyrazine thiosemicarbazone) was prepared via a facile hydrothermal method and successfully characterized. Compound **1** exhibited high stability in a wide range of pH values from 4 to 10. **1** demonstrated significant antibacterial activity, with minimum inhibitory concentration (MIC) values in the range of 0.0019–0.2400 µg/mL against four types of bacteria, including *Staphylococcus aureus* (*S. aureus*), *Bacillus subtilis* (*B. subtilis*), *Escherichia coli* (*E. coli*), and *Agrobacterium tumefaciens* (*A. tumefaciens*). Further time-kill studies indicated that **1** killed almost all (99.9%) of *E. coli* and *S. aureus.* Meanwhile, the possible antibacterial mechanism was explored, and the results indicate that the antibacterial properties of **1** originate from the synergistic effect between [Ni(L)(HL)]^+^ and [PMo_12_O_40_]^3−^. In addition, **1** presented effective adsorption of basic fuchsin (BF) dyes. The kinetic data fitted a pseudo-second-order kinetic model well, and the maximum adsorption efficiency for the BF dyes (29.81 mg/g) was determined by the data fit of the Freundlich isotherm model. The results show that BF adsorption was dominated by both chemical adsorption and multilayer adsorption. This work provides evidence that **1** has potential to effectively remove dyes and pathogenic bacteria from wastewater.

## 1. Introduction

In recent years, industrialization and urbanization have deteriorated water quality, resulting in a higher number of pollutants in the water, such as pathogenic bacteria, organic pollutants, and heavy metal ions [1,2,3]. Pathogenic bacteria are one of the most common contaminants in wastewater. Meanwhile, the widespread use of antibiotics not only leads to an increase in bacterial resistance but may also result in the emergence of super bacteria, thereby posing a threat to global public safety and the ecological environment [4,5]. For the last few years, a tremendous amount of effort has been devoted to the design of antimicrobial materials [6,7,8,9]. At the same time, significant amounts of co-products such as various organic dyes have been released into water due to the rapid development of the dyeing and printing industry, causing serious industrial wastewater contamination globally. Therefore, there is an urgent demand for the removal of harmful dyes from polluted water to alleviate environmental issues. To date, many techniques and methods have been utilized for the removal of toxic dyes, including adsorption, photocatalysis, and biodegradation [10]. Remarkably, the adsorption method has been extensively applied for the treatment of dye wastewater because of its low cost, simple operation, and practicality [11,12,13]. It is essential to remove contaminants from water before releasing it into the environment. Thus, it is urgent to construct multifunctional materials with both antibacterial performance and dye adsorption properties [14].

Polyoxometalates (POMs) are unique nanoscale polyanionic clusters comprising early transition metal oxides that have potential application in biological and medical areas, material science, and water treatment [15,16]. Given their adjustable structure and extensive physiochemical properties, considerable efforts and studies have been devoted to the construction of POM-based hybrids and nanocomposites with antimicrobial and adsorptive properties. In 2018, Rompel’s group reported the antibacterial activity of 29 different POMs against *Moraxella catarrhalis*, and the Preyssler-type polyoxotungstate [NaP_5_W_30_O_110_]^14−^ demonstrated excellent antibacterial activity with an MIC of 1 µg/mL [17]. In 2020, our group reported two Keggin-type Co-POMs with significant antibacterial properties, namely [Co(L)_2_]_3_[PMo_12_O_40_] and [Co(L)_2_]_3_[PW_12_O_40_] [18]. In 2021, an aluminum-substituted Keggin [Al(H_2_O)GeW_11_]^4−^ was synthesized by Rompel’s group, and the Al-POM indicated enhanced antibacterial activity compared with simple aluminum salt against *Moraxella catarrhalis* (MIC = 4 µg/mL) and Gram-positive *Enterococcus faecalis* (MIC = 128 µg/mL), respectively [19]. Obviously, both polyoxomolybdates and polyoxotungstates exhibit excellent antibacterial activity. In addition, POM-based composites also exhibit adsorption properties for dye removal. Dong’s group presented a feasible and versatile approach to introduce POMs to construct a functionalized membrane via a novel sol-gel method, and this material effectively removed reactive black 5 dyes [20]. In 2020, Mardiyanto’s group investigated an Ni-Fe layered double hydroxide (LDH) intercalated by Keggin-type POM [SiW_12_O_40_]^4−^, and the material showed a large adsorption capacity for malachite green (MG) [21]. However, little information is available on POMs with both antimicrobial and adsorption properties. Hence, the design and construction of POMs-based materials are vital to realizing bifunctionalization of such materials.

POM-based inorganic-organic hybrids exhibit enhanced antimicrobial and adsorption properties compared to pure POMs [22,23]. Among the organic ligands, 2-acetylpyrazine thiosemicarbazone (HL) showed excellent antibacterial activity in pharmacological applications, being a superior candidate for the construction of POM-based inorganic-organic hybrids [24,25,26,27]. This study aimed to construct a multifunctional POM compound with both antibacterial performance and dye adsorption properties. An Ni-based polyoxomolybdate compound [Ni(L)(HL)]_2_H[PMo_12_O_40_] 4H_2_O (**1**) was constructed using the facile hydrothermal method and systematically characterized. **1** not only showed excellent antibacterial activity but also exhibited adsorption properties for BF dyes, which supports the use of POMs as water purification materials in the killing of microorganisms and removal of organic contaminants synchronously. The antibacterial activities and adsorption performance of **1** were explored in detail. Furthermore, the possible antibacterial and adsorption mechanisms of **1** were also proposed on the basis of the experimental results.

## 2. Results and Discussion

X-ray single-crystal diffraction analysis shows that **1** crystallizes in the space group P-1 [28,29], as shown in Appendix A. The molecular structure of **1** contains two [Ni(L)(HL)]^+^ cations, one H^+^, one Keggin [PMo_12_O_40_]^3−^ anion, and four lattice water molecules (Figure 1A). The coordination configuration of the Ni(II) ions was calculated using the SHAPE 2.1 program [28], as shown in Appendix A, and the result indicates that the Ni(II) centers are located in a distorted octahedron geometry. In **1**, each six-coordinate Ni(II) ion is surrounded by four N atoms and two S atoms from two 2-acetylpyrazine thiosemicarbazone ligands (Ni–N: 2.026–2.102 Å, Ni–S: 2.391–2.398 Å, ∠S–Ni–S: 93.88 (2)°, ∠N–Ni–S: 83.2 (1)–160.9 (1)°, ∠N–Ni–S: 77.9 (1)–172.3 (1)°). Meanwhile, the two dihedral angles of (Ni1–S1–C1–N2–N3)⋯(Ni1–N3–C2–C4–N5) and (Ni1–N10–C11–C9–N8)⋯(Ni1–S2–C8–N7–N8) were found to be about 2.436° and 0.659°, respectively. Obviously, each HL ligand includes one imine and one pyrazine group, which contribute to the formation of hydrogen bonds, and five kinds of H-bond interactions (N7–H7⋯O1, N6–H6A⋯O1, N6–H6B⋯O5, N1–H1A⋯O1W, N1–H1A⋯O10) were observed in the structure of **1**. The [Ni(L)(HL)]^+^ cations and [PMo_12_O_40_]^3−^ polyoxoanions are connected by hydrogen bonding (Figure 1B), with further stacking to form the layered structure (Appendix A).

FT-IR spectroscopy was employed to identify the functional groups of **1** (Figure 2A). The peaks at 3174 cm^−1^ and in the region of 1608–1395 cm^−1^ belong to the stretching vibration of ν(N–H) and ν(C=N) of HL, and the (P–Oa) vibrations of **1** can be observed at 1158–1057 cm^−1^ [30]. Additionally, three characteristic vibration bands of **1** at 953, 840, and 786 cm^−1^ are assigned to the asymmetric stretching vibrations of ν(Mo–O_d_), ν(Mo–O_b_–Mo), and ν(Mo–O_c_–Mo) [31], respectively. The characteristic absorption bands prove the existence of the [PMo_12_O_40_]^3−^ anion in the structure of **1**. In addition, the Raman spectrum (Figure 2B) of **1** shows main characteristic vibration bands at 814, 883, and 935 cm^−1^, which correspond to the symmetrical stretching Mo–O–Mo, Mo=O, and P–O [32,33], respectively. Hence, it can be concluded that the Keggin-type POM compound was successfully prepared.

The X-ray diffraction (XRD) spectra were used to confirm the phase purity of **1**. The XRD patterns for **1** from the experiments are in accordance with the simulation results (Figure 2C).

To evaluate the stability of **1**, the ultraviolet–visible (UV-vis) spectroscopy at different pH values from 2 to 12 was investigated in detail. The spectra of **1** indicate three characteristic bands in the region of 200–500 nm. The bands at 230 and 307 nm are attributed to the charge transfer transition of Od→Mo and Ob, Oc→Mo bands [34], respectively. Meanwhile, weak absorption is seen at 411 nm, which may be due to the ligand to metal charge transfer (LMCT) transitions [35]. When pH value is 3.8, the band strength at 411 nm becomes weak. The peak gradually flattens as the pH value decreases (Figure 2D). In addition, a weak band at 230 nm is observed at pH 9.9 (Appendix A), and the peak disappears as the pH value increases, indicating that the POM framework was decomposing. Based on these experimental results, **1** possesses good stability (pH = 4–10).

The results of the X-ray photoelectron spectroscopy (XPS) analysis demonstrate the detailed chemical compositions of the synthetic compound and the valence state of these elements. Figure 3A shows that the elements of Ni, O, N, C, P, Mo, and S exist in **1**. The XPS spectrum of Ni 2p suggests two obvious peaks at the binding energies of 871.7 and 854.2 eV (Figure 3B), which are assigned to Ni^2+^ 2p1/2 and Ni^2+^ 2p3/2, respectively [36]. Additionally, the presence of two satellite peaks in the Ni 2p XPS spectra further demonstrates the existence of octahedrally coordinated Ni (II) ions [37,38].

Scanning electron microscopy (SEM) images were used to investigate the morphology of **1**. **1** has a quadrilateral structure with a flat surface and most crystal compounds show a similar shape (Figure 4A,B). Furthermore, **1** showed a range of widths between 50 and 86 µm and lengths between 95 and 190 µm, respectively. Additionally, the SEM elemental mappings of Ni (green), Mo (red), and P (yellow) are observed in the same micrometer region and these components are uniformly scattered in **1**. The elemental mappings show the color intensity of the components, confirming that the content of the Mo element is higher than that of the Ni and P elements (Figure 4C–E), which is in agreement with the ICP analysis.

### 2.1. Antibacterial Properties

In this study, the antibacterial activities were assessed using four bacterial strains (*E. coli*, *A. tumefaciens*, *S. aureus*, and *B. subtilis*) and the disc diffusion method to confirm the MIC values. The inhibition zone diameters of **1** against *E. coli* bacteria and *B. subtilis* bacteria were measured (Appendix A). **1** showed high antibacterial activities against both *E. coli* bacteria and *B. subtilus* bacteria, with two similar inhibition zone diameters of 20 mm at a concentration of 4 mg/mL. As the concentrations decreased, the inhibition zone diameters diminished little by little. To explore the antibacterial efficiency of the different components, the MIC values were explored, including HL, Ni(ClO_4_)_2_·6H_2_O, H_3_[PMo_12_O_40_] (Pmo_12_), **1**, two reference medicines, amoxicillin trihydrate (Am), and kanamycin sulfate (Kan), as illustrated in Table 1. It can be seen that a single component nickel ion exhibited inferior antibacterial activity, with a high MIC value. In comparison, ligand HL and H_3_[PMo_12_O_40_] demonstrated lower MIC values, indicating that they have higher antibacterial activity than nickel salts. It is worth noting that the MIC values of **1** were 0.015 and 0.24 µg/mL against the Gram-negative bacteria *E. coli* and *A. tumefacies* and 0.0019 and 0.03 µg/mL against the Gram-positive bacteria *S. aureus* and *B. subtilis*, respectively, which demonstrates **1** exhibited high-efficiency antibacterial activities. At the same time, compared with Am and Kan, **1** possessed a lower MIC value, indicating that **1** may be used as a potential antibacterial agent.

### 2.2. Time-Kill Studies

In order to further evaluate the bactericidal or bacteriostatic ability of **1**, time-kill studies were carried out using the viable cell count method (VCC). The numbers of *E. coli* bacterium and *S. aureus* bacterium declined as time increased (Figure 5A,B). After being exposed to **1** for 4 h, about 81% of the *E. coli* bacterium were killed while the killing rate of *S. aureus* bacterium reached 97%, indicating that **1** could kill *S. aureus* bacterium quickly. As expected, both types of bacteria were nearly killed within 6 h after treatment with **1**. In addition, it is concluded that **1** differs in its sensitivity to *S. aureus* and *E. coli* bacteria. Meanwhile, the experimental results were also analyzed by statistical analysis (Appendix A), indicating that **1** showed significant antibacterial activity against *S. aureus* and *E. coli* bacteria.

### 2.3. Adsorption Performances

To evaluate whether the adsorption capacity of **1** was affected by different factors, adsorption experiments were performed. The size range of the solid crystals was 50–200 µm in the experiments. Some influencing factors (dye concentrations, adsorption dosage, different dyes and pH values) were considered in the experiments. To explore the optimal mass concentrations of the BF dyes, **1** (20 mg) was added to 50 mL of aqueous solutions of BF at room temperature with different mass concentrations of 5, 10, and 15 mg L^−1^. It can be seen that the content of the BF dyes showed an obvious decrement (71%) of 15 mg L^−1^ in the absorption spectra (Appendix A), which signifies that **1** exhibited a better adsorption performance. However, **1** removed about 61% and 42% of the BF dyes at the BF concentrations of 5 and 10 mg/L, respectively. Additionally, it should be mentioned that the dye concentration decreased rapidly during the first 30 min, which may be attributed to a large number of active sites of the dye being occupied via electrostatic interaction [10], then gradually reduced after 30 min. By analyzing the C/C_0_ versus BF concentration and contact time, the C/C_0_ value decreased with the increase in the contact time until the maximum adsorption was observed at 90 min for all concentrations of the BF dyes. Finally, the adsorption reached equilibrium and there was almost no further increase in the adsorption. Commonly, the ratio between the initial number of BF molecules and the available surface area is large at higher concentrations. Thereby, fractional adsorption relies on the initial concentration of BF dyes. However, fewer sites of adsorption are available at lower concentrations, thus dye removal depends on the concentration of dyes [39]. According to these results, it appears that the adsorption uptake is sharp for the first 30 min, after which it goes through a slower adsorption process, thereby obtaining saturation at 90 min [27,39]. The whole adsorption step is divided into two processes, including the first fast-speed surface adsorption and the slower diffusion adsorption inside the particles during the second period.

The adsorption dosage also plays an important role in the removal of organic dyes. **1** showed a distinct adsorption capacity of 0.2, 0.4, and 0.8 mg/mL with a removal ratio of 69.6%, 79.9%, and 67.2% in the dark for 90 min (Figure 6A). Moreover, according to further simulation and analysis of the data, the maximum adsorption capacity of the above adsorption dosages was 26.41, 29.81, and 25.85 mg/g, respectively. The adsorption ability of **1** decreased with an increase in the adsorbent concentration from 0.4 to 0.8 mg/mL. This may have been caused by a reduction in the specific surface area of POM and available adsorption sites for BF dyes due to the enhanced repulsive interaction and agglomeration between POM particles [40,41]. In addition, to further verify this conclusion, the adsorbent concentration (1 mg/mL) was utilized to further explore the adsorption capacity of **1**, and the adsorption rate was 62.07% according to the removal efficiency formula (Equation (S2)). The results demonstrated that the optimal concentration of **1** was 0.4 mg/mL for BF adsorption.

Additionally, the adsorption ability of **1** for the other organic dyes was also investigated under dark conditions at room temperature. **1** showed a selective adsorption capacity for the methylene blue (MB), gentian violet (GV), and BF dyes, with removal ratios of 46.6%, 69%, and 90.1% (Appendix A), respectively. However, the removal ratio of methyl orange (MO) was close to 0, which is consistent with the result of MO dye removal in a previous report [42,43]. As a result, **1** can efficiently and selectively remove cationic dyes from wastewater.

The adsorption performances of **1** for the BF dyes were tested under acidic, alkaline, and neutral solution environments since the pH values of natural wastewater varied. As shown in Appendix A, **1** showed the best adsorption performance in the neutral solution and removed about 41.8% of the BF dyes through adsorption. However, **1** showed almost no adsorption for BF dyes under the acidic and alkaline circumstances, which is also consistent with previous reports [10,44].

To confirm the stability of **1** before and after BF adsorption, a series of tests were executed. The SEM test results indicate that the morphology of **1** did not change after adsorption, and the test result of the element mapping reveal that the Ni, Mo, and P elements existed in **1** after adsorption. Furthermore, the peak positions from the IR and Raman spectrum were nearly constant before and after adsorption, which indicates that the functional groups of **1** did not change. The above test results confirm that the structure of **1** remained after adsorption. The SEM test of **1** after adsorption (Appendix A) and the IR and Raman spectrum of **1** after adsorption Appendix A.

For practical applications of adsorbents, it is vital to explore their stability and regeneration ability during the adsorption process. The adsorbent **1** could be regenerated by simple dry treatment at 60 °C for 180 min. **1** retained a high removal efficiency over three cycles. As depicted in Appendix A, the adsorption capacities decreased for each new cycle after dry treatment. The original adsorption capacity of **1** for the BF dyes was 29.81 mg/g, which then dropped to 24.45 mg/g after three cycles. Consequently, **1** showed fine stability and regeneration for BF adsorption.

The UV–vis spectra of the dye solution at 330-min intervals are presented in Appendix A. After 330 min of adsorption, the absorption peaks of the wavelength at 542 nm decreased continuously and the dye solution became almost colorless, indicating that most of the BF dyes in water were removed. The cationic dye adsorption is possibly attributed to electrostatic interaction, *π*⋯*π* interactions, hydrophobic interaction, and hydrogen bond [20,41].

#### 2.3.1. Adsorption Kinetics

By fitting the experimental data to the three models (Figure 6B–D), we report the kinetic parameters of BF adsorbed onto **1** (Table 2). It can be seen that the pseudo-second-order rate expression (R^2^ > 0.99) is in good agreement with the experimental data for BF. The higher coefficients of the pseudo-second-order model show that the adsorption process of **1** for the BF dyes was mainly governed by chemisorption. Moreover, the adsorption rate was found to be slow due to the occupied active sites on the surface of **1** as the solution concentration increased.

#### 2.3.2. Equilibrium Results

The equilibrium curves (Figure 7) of **1** for the BF dyes were fitted with the Langmuir, Freundlich, Dubinin–Radushkevich, and Temkin models, and the fitting parameters are displayed in Table 3. It shows that the four used models presented different values of R^2^. Moreover, the Freundlich model presented a higher value than the Dubinin–Radushkevich, Temkin, and Langmuir models; thus, it can be assumed that the BF molecules were not distributed equally. The maximum adsorption capacity of **1** for the BF dyes was 29.81 mg/g.

### 2.4. Probable Action Mechanisms

The Gram-negative bacteria *E. coli* possesses a dense lipopolysaccharide layer (outer membrane), containing lipid, oligosaccharide, and polysaccharide molecules whereas *S. aureus*, a type of Gram-positive bacterium, has a thicker peptidoglycan layer without any outer membrane [45]. Moreover, the outer membrane provides the cell with a strong permeability barrier, which can resist macromolecules and hydrophobic molecules [18]. Hence, to better understand the antibacterial mechanism of **1**, *E. coli* was selected as a model to study the effect of **1**. In general, the negatively charged bacterial cell wall provides antibacterial materials with coordination sites, facilitating the interaction between the membrane and compound. Thus, if the balance of the potential on the bacterial cell surface is disrupted, which in turn changes the permeability and integrity of the cells, the destruction of the cell wall/membrane can cause an outflow of intracellular components, leading to cell death [46,47].

To intuitively observe the interaction of cells with **1**, SEM images were used to observe the morphological changes in the *E. coli* cells. The untreated *E. coli* cells exhibited a natural rod-shaped surface and smooth appearance with complete cell walls/membranes (Figure 8A). Comparatively, obvious morphological and physical damage of cells was observed after treatment with **1** (Figure 8B), including severe perforation and disturbance of the cell membranes. The SEM image results indicate that **1** led to serious damage to the bacterial morphology, further damaging the integrity of the bacteria, which may be one of the primary antibacterial mechanisms of **1**. Thus, we propose that the electrostatic interaction promoted **1** to capture the negatively charged bacteria cell wall, which led to severe destruction of the bacterial cells, induced intracellular components’ leakage, and interfered with the bacteria’s normal physiological activity, thereby leading to bacterial cell death.

It is generally assumed that antibacterial materials generally react with the bacterial cell wall/membrane, resulting in the leakage of intracellular substances such as K^+^, DNA, and RNA, thus inducing damage to the cell membrane structures and disruption of the cell membrane integrity [48,49]. Significant changes were clearly observed in the experimental group (Figure 9A). Compared to the data changes (0.0913 ± 0.005) in the control group, the optical density of the 260 nm mensuration with the compound test group (0.2074 ± 0.0004) was obviously higher than the control group, indicating that **1** could damage the bacterial cell membrane. Meanwhile, the disruption of the bacterial cell and the leakage of cytoplasm induced by **1** were further proven by quantitative detection of the leakage of intracellular proteins, which is regarded as another momentous indicator of cytoplasmic leakage. The protein content in the test group in the presence of **1** was obviously higher than the control group (Appendix A). High leakage of ribonucleic acid (RNA) and protein showed that **1** has significant antibacterial activity, enhancing the penetrability of the bacterial cell membrane and resulting in more leakage of intracellular RNA and proteins. Furthermore, the activity of the bacterial respiratory chains was tested to evaluate the probable antibacterial mechanism. The respiration chain dehydrogenase activity was significantly reduced in the test group after the treatment with **1** (Figure 9D), which indicates that the normal metabolism of bacteria was inhibited. Therefore, it can be inferred that **1** can break through the extracellular membrane and the barrier of the cytoplasm, destroying the respiratory chain dehydrogenase of bacterial cells and inhibiting the respiration of bacterial cells.

Apart from damaging the integrity of the bacterial structure via the direct physical contact between **1** and bacterial cells, oxidative stress is another significant antibacterial mechanism. Nitro blue tetrazolium (NBT) can be reduced to blue-violet formaldehyde in the presence of O^2−^. Therefore, the index of the reactive oxygen species (ROS) content was determined by the production of blue-violet formaldehyde [50,51,52]. According to the previous literature, a high ROS content can lead to oxidative stress in bacteria cells [6,43], resulting in disruption of the bacterial cell membrane, intracellular substances, and so on. **1** induced an obviously higher level of ROS production than the control group (Figure 9B). Glutathione (GSH) can prevent cell damage caused by oxidative stress and is involved in protecting cells. However, antibacterial materials can convert sulfhydryl groups (-SH) of GSH to disulfide bonds (-S-S) to induce the oxidized process, and the content of -SH groups in GSH can be recognized by the Ellman method [53]. The data on the GSH content in the negative control group was almost unchanged (Figure 9C), meaning that the experimental conditions would not result in GSH oxidation. Meanwhile, the loss of GSH in the positive control group (H_2_O_2_) was about 67.1%, which is consistent with a previous literature report [46]. In contrast, the extent of GSH oxidation was reduced after the addition of **1**, and the oxidation ratio reached 34.6% after 15 min. The results showed that **1** may be released into the interior of the bacterial cell by the injured cell membrane or penetrate bacteria cells. Consequently, the oxidative stress response may be responsible for the bacterial cell apoptosis.

Specifically, there are several factors involved in bacterial cell death, including cell wall/membrane rupture, leakage of intracellular components, and the oxidative stress response. Based on our studies and previous literature [47,54], POMs can interact with the bacterial cell wall/membrane and destroy the integrity of the cell membrane, allowing intracellular substances to leak and disturb the normal cellular metabolism. In detail, the possible antibacterial action mechanisms of POMs are provided, as follows: (1) POMs interact with the bacterial cell wall, further localizing within or at the inner membrane of bacteria; (2) POMs are able to interfere with proteins that are responsible for some bacteria [23]; (3) as most POMs are highly redox active, they could impair the bacterial respiratory system by oxidizing some important electron carriers and thus affect ATP production [55]; and (4) POMs can also produce and elevate ROS by oxidizing proteins, lipids, GSH, and other bacterial compounds, which leads to the depletion of the GSH pool [56]. Based on the above analysis, it is concluded that bacterial death is mainly due to two points between membrane disruption and the oxidative stress process. On the one hand, [Ni(L)(HL)]^+^ can interact with negatively charged bacterial cell membranes. On the other hand, the oxidizing [PMo_12_O_40_]^3−^ anions may harm the bacterial respiratory system by oxidizing some important electron carriers [], thereby affecting ATP production. These two reasons cause the bacterial cell membrane to break down. Moreover, [PMo_12_O_40_]^3−^ anions can produce ROS by oxidizing GSH, leading to the depletion of the GSH pool. The synergistic interactions of [Ni(L)(HL)]^+^ and [PMo_12_O_40_]^3−^ in **1** resulted in membrane disruption and the oxidative stress process. The proposed action mechanism of **1** against *E. coli* is presented in Figure 1.

## 3. Materials and Methods

### 3.1. Materials

Sodium molybdate dihydrate (Na_2_MoO_4_·2H_2_O, 99%), nickel perchlorate hexahydrate (Ni(ClO_4_)_2_·6H_2_O, 99%), methyl alcohol (CH_3_OH), acetonitrile (CH_3_CN), and ethylenediaminetetrakis (methylenephosphonic acid) (EDTMP, 98%) were available from Beijing Innochem Science & Technology Co., Ltd. (Beijing, China). Aqueous solutions of BF, MB, MO, and GV were obtained from J&K Scientific Ltd. (Beijing, China) and prepared using deionized water. Gram-negative bacteria (*E. coli*, *A. tumefaciens*) and Gram-positive bacteria (*S. aureus*, *B. subtilis*) were purchased from the China General Microbiological Culture Collection Center (Beijing, China).

### 3.2. Preparation of ***1***

The ligand HL was synthesized according to the literature [26]. **1** was synthesized as follows: Ni(ClO_4_)_2_·6H_2_O (0.475 g, 1.3 mmol), Na_2_MoO_4_·2H_2_O (0.48 g, 2.0 mmol), EDTMP(0.35 g, 0.8 mmol), and HL (0.097 g, 0.5 mmol) were added to 60 mL of solvent with acetonitrile and water (1:2) and stirred continuously for 40 min at room temperature, which was followed by the use of NaOH (1 M) and HCl (1 M) solution to adjust pH to 3. Thereafter, the above mixture solution was tardily transferred to a 30-mL Teflon-lined autoclave, where such compound was heated at 110 °C for 72 h and subsequently cooled to room temperature. After filtration and washing with distilled water three times, the black quadrilateral crystals of **1** were obtained, and finally dried at 60 °C for 12 h (yield: 0.252 g, 72.2%, based on HL). EA and ICP (%): calcd for C 11.91, H 1.57, N 9.92, Ni 4.16, Mo 40.76 (%). Found for C 11.82, H 1.52, N 9.94, Ni 4.09, Mo 40.66 (%).

X-ray crystallography, antibacterial activity test, time-dynamic bactericidal test, adsorption performance of **1**, and antibacterial action investigation and characterization are provided in the Appendix A. The crystal parameters, structure refinement data, and CCDC reference number (2119325) for **1** are listed in Appendix A.

## 4. Conclusions

In conclusion, an Ni-based Keggin POM derived from [PMo_12_O_40_]^3−^ and thiosemicarbazone was successfully synthesized using a facile hydrothermal method and systematically characterized. **1** revealed significant antibacterial activity against *E. coli*, with MIC values of 0.015 µg/mL and minor MIC values of 0.0019 µg/mL against *S. aureus*. Moreover, time-kill studies demonstrated the total bactericidal effect of **1** on the particular bacterium, revealing that the killing rate against S. aureus bacterium reached 97% after 4 h and almost 100% after 6 h. Further potential antimicrobial mechanisms were also investigated using the quantitative assay method. The synergistic interactions of [Ni(L)(HL)]^+^ and [PMo_12_O_40_]^3−^ in **1** resulted in membrane disruption and the oxidative stress response in the antimicrobial process. In addition, **1** showed significant activity in the removal of BF dyes (79.9%) within 90 min in wastewater. The pseudo-second-order kinetic model and the Freundlich isotherm model can properly describe the adsorption process. The BF adsorption is mainly governed by chemisorption and multilayer adsorption. This work displays more possibilities of POMs as multifunctional materials with antibacterial performance and dye adsorption properties for their application in water pollution treatment.

## Data Availability

Not applicable.

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
