# Peer review of "A Nickel-Containing Polyoxomolybdate as an Efficient Antibacterial Agent for Water Treatment"

_ijms, 2022, doi:10.3390/ijms23179651_

Round 1

Reviewer 1 Report

Comments to the Author

In this study ‘’A nickel-containing polyoxomolybdate as an efficient antibac-2 terial agent for water treatment’’

The article is written well, however, before publication will need more work.

Major corrections:

1. Insufficient characterization techniques. The author should add EDX and TEM of prepared polyoxomolybdate before and after treatment/adsorption to verify the structural changes in the complex.

2. Explain in detail the specific mechanism of antibacterial action of polyoxomolybdate complex.

3. The author should measure the recyclability study of polyoxomolybdate complex for basic fuchsin dye adsorption.

Minor corrections:

1. There are few typographical and grammatical errors in the manuscript. Hence, the manuscript should be carefully checked and necessary corrections should be done. Like ‘’In addition, 1 presented effective adsorption for basic fuchsin (BF) dyes.’’ Dye or dyes?

2. The purpose and novelty of the research should be added in the Introduction section.

3. ‘’Following the preparation of the original concentration of 1 (1 mg/mL), different concentrations were obtained by two-fold dilution in this assay’’. Which solvent is used for the preparation of dilution?

4. The adsorption of BF dye after first 30 min is too high and then become almost constant. Why?

5. ‘’Distinct adsorption capacity of 0.2 mg/mL, 0.4 mg/mL and 0.8 mg/mL with the 199 removal ratio of 69.6 %, 79.9 %, and 67.2 % in the dark of 90 min.’’ Why the adsorption ability decreases with increasing adsorbent concentration from 0.4 mg/mL to 0.8 mg/mL?

Reviewer 2 Report

In my opinion the paper with title: “

 A nickel-containing polyoxomolybdate as an efficient antibacterial agent for water treatment should be published in the International Journal of the Molecular Sciences if the authors will make the following corrections and specifications.

Please enter the meaning of all abbreviations used at their first mention in the text.

 For example, MIC – Chapter-Introduction, LMCT (line 122), MB, GV, line 207, etc.

Please write the name of the microorganism species in the same way in the text (usually Italic).

Please enter the name of the parameter that varies in the legend, on the figures

For example, in figure 2 D., the meaning of the numbers that are mentioned in the Legend (pH) is not presented

In accordance with the Instructions for Authors in the paper at the Chapter Materials and Methods: “They should be described with sufficient detail to allow others to replicate and build on published results. New methods and protocols should be described in detail while well-established methods can be briefly described and appropriately cited. Present the name and version of any software used and make clear whether computer code used is available. Include any pre-registration codes

The Materials and Methods chapter is very poorly represented in this article.

Please present a detailed description of the working method and working conditions for each type of experiment,

Please specify the mathematical equations used and the meaning of the parameters to describe the studied processes.

Please specify the type and manufacturer of the equipment and reagents used, etc.

Also, you should present the calculation relationships for different sizes whose variation was represented or mentioned in the text, for example for the degree of dye removal

Please mention the size range of the solid crystals used in the experiments.

Also, mention how to ensure a certain pH value.

What is the working method and the working conditions through which the stability of the crystals was highlighted in various conditions?

Round 2

Reviewer 1 Report

I appreciate the author's efforts. The manuscript is improved well and I will recommend after minor revision with the following comments.

1.         Comments No 2 explanations of antimicrobial mechanism should be included in the manuscript or supplementary file.

2.         The Recyclability / reusability study should be included in the manuscript or supplementary file.

3.         There are total 50 references, while I found 48 references in the main text.

Author Response

Henan Key Laboratory of Polyoxometalates

College of Chemistry and Chemical Engineering

Henan University

Kaifeng, 475004

  1. R. China

Fax: +86-371-23881589.

E-mail: mpt@henu.edu.cn

Aug, 18 2022

Manuscript ID: ijms-1835452

Title: A nickel-containing polyoxomolybdate as an efficient antibacterial agent for water treatment

Dear reviewers,

First of all, I am very grateful to you for your time and effort you expend on our manuscript as well as your patience and great generosity to give us the opportunity to revise our manuscript. Secondly, we deeply thank you for your valuable and helpful suggestions. Thirdly, in light of your comments, we have tried our best to make appropriate changes in the manuscript point by point (highlighted in yellow). The details are listed as follows:

To the comments of Reviewer #1:

  1. Comments No 2 explanations of antimicrobial mechanism should be included in the manuscript or supplementary file.

Response: Thank you very much for your good suggestions. The explanations of antimicrobial mechanism has been added in the revised manuscript (Page 11, lines 372-390).

  1. The Recyclability / reusability study should be included in the manuscript or supplementary file.

Response: Thank you very much for your good suggestions. The reusability study has been added in the revised manuscript (Page 7, lines 254-260).

  1. There are total 50 references, while I found 48 references in the main text.

Response: Thank you very much for your good suggestions. The references has been revised in the revised manuscript.

If you have any questions, please feel free to contact us.

Looking forward to your favorable reply. Thank you very much.

Sincerely,

Pengtao Ma, Lin Sun

mpt@henu.edu.cn, sunlin@vip.henu.edu.cn

Reviewer 2 Report

In my opinion the paper with title: “

 A nickel-containing polyoxomolybdate as an efficient antibacterial agent for water treatment should be published in the International Journal of the Molecular Sciences if the authors will make the following aditions:

In accordance with the Instructions for Authors in the paper at the Chapter Materials and Methods: “They should be described with sufficient detail to allow others to replicate and build on published results. New methods and protocols should be described in detail while well-established methods can be briefly described and appropriately cited. Present the name and version of any software used and make clear whether computer code used is available. Include any pre-registration codes

The Materials and Methods chapter must be completed with a detailed description of the working methods and working conditions for each type of developed experiment.

Author Response

Henan Key Laboratory of Polyoxometalates

College of Chemistry and Chemical Engineering

Henan University

Kaifeng, 475004

  1. R. China

Fax: +86-371-23881589.

E-mail: mpt@henu.edu.cn

Aug, 18 2022

Manuscript ID: ijms-1835452

Title: A nickel-containing polyoxomolybdate as an efficient antibacterial agent for water treatment

Dear reviewers,

First of all, I am very grateful to you for your time and effort you expend on our manuscript as well as your patience and great generosity to give us the opportunity to revise our manuscript. Secondly, we deeply thank you for your valuable and helpful suggestions. Thirdly, in light of your comments, we have tried our best to make appropriate changes in the manuscript point by point (highlighted in yellow). The details are listed as follows:

To the comments of Reviewer #2:

The Materials and Methods chapter must be completed with a detailed description of the working methods and working conditions for each type of developed experiment.

Response: Thank you very much for your good suggestions. The Materials and Methods chapter has been adjusted and modified in the revised supporting information. We have added a paragraph to the main text (Page 12, lines 416-419), as follows: X-ray crystallography, antibacterial activity test, time-dynamic bactericidal test, adsorption performance of 1, antibacterial action investigation and characterization are provided in the supporting information. The crystal parameters, structure refinement data and CCDC reference number (2119325) for 1 are listed in Table S1.

If you have any questions, please feel free to contact us.

Looking forward to your favorable reply. Thank you very much.

Sincerely,

Pengtao Ma, Lin Sun

mpt@henu.edu.cn, sunlin@vip.henu.edu.cn